# Vertically Aligned NiCo_2_O_4_ Nanosheet-Encapsulated Carbon Fibers as a Self-Supported Electrode for Superior Li^+^ Storage Performance

**DOI:** 10.3390/nano9091336

**Published:** 2019-09-18

**Authors:** Yongchao Liu, Jintian Jiang, Yanyan Yuan, Qinglong Jiang, Chao Yan

**Affiliations:** 1School of Materials Science and Engineering, Jiangsu University of Science and Technology, Zhenjiang 212003, China; yongchao.liu@foxmail.com (Y.L.); jiangjintianzzz@outlook.com (J.J.); yuan.yanyan@just.edu.cn (Y.Y.); 2School of Materials Science and Engineering, Hefei University of Technology, Hefei 230009, China; 3Department of Chemistry and Physics, University of Arkansas, Pine Bluff, AR 71601, USA

**Keywords:** binary transition metal oxide, NiCo_2_O_4_ nanosheets, biomass-derived carbon fibers, self-supported electrode

## Abstract

Binary transition metal oxides (BTMOs) have been explored as promising candidates in rechargeable lithium-ion battery (LIB) anodes due to their high specific capacity and environmental benignity. Herein, 2D ultrathin NiCo_2_O_4_ nanosheets vertically grown on a biomass-derived carbon fiber substrate (NCO NSs/BCFs) were obtained by a facile synthetic strategy. The BCF substrate has superior flexibility and mechanical strength and thus not only offers a good support to NCO NSs/BCFs composites, but also provides high-speed paths for electron transport. Furthermore, 2D NiCo_2_O_4_ nanosheets grown vertically present a large contact area between the electrode and the electrolyte, which shortens the ions/electrons transport distance. The nanosheets structure can effectively limit the volume change derived from Li^+^ insertion and extraction, thus improving the stability of the electrode material. Therefore, the synthesized self-supporting NCO NSs/BCFs electrode displays excellent electrochemical performance, such as a large reversible capacity of 1128 mA·h·g^−1^ after 80 cycles at a current density of 100 mA·g^−1^ and a good rate capability of 818.5 mA·h·g^−1^ at 1000 mA·g^−1^. Undoubtedly, the cheap biomass carbon source and facile synthesis strategy here described can be extended to other composite materials for high-performance energy-storage and conversion devices.

## 1. Introduction

The use of renewable clean energy has promoted the development of energy storage systems (ESSs), especially rechargeable lithium-ion batteries (LIBs), which are widely used in smart appliances, medical devices, and urban buses due to their long life cycle, high power density, and environmental friendliness [1,2,3]. Transition metal oxides (TMOs), such as Co_3_O_4_, Fe_2_O_3_, NiO, and MnO_2_, have been extensively studied as promising anode materials for LIBs, having more than twice the capacity of traditional graphite anodes [4,5,6,7,8]. However, the disadvantages of poor stability, low electrical conductivity, and large volume expansion hinder their applications. Binary transition metal oxides (BTMOs), especially nickel cobaltate (NiCo_2_O_4_), have been conceived as promising cost-effective and scalable electrode materials for energy storage due to their inherent advantages, including low cost, abundant resources, and environmental benignity [9,10,11]. More importantly, the binary transition metal oxide NiCo_2_O_4_ possesses much better electrical conductivity and higher electrochemical activity than Co_3_O_4_ and NiO [12,13,14].

Unfortunately, NiCo_2_O_4_ has poor structural stability during the charge–discharge process, similar to other transition metal oxide materials, and it detaches easily from the current collector resulting in a large amount of irreversible capacity loss. Furthermore, active materials (such as NiCo_2_O_4_) require a conductive agent and a binder when used in conventional electrodes, which inevitably decreases the overall storage capacity of the electrode. Therefore, some strategies were designed to prepare self-supporting electrodes. For example, NiCo_2_O_4_ was grown directly on the metal conductive substrates (Ti foil, nickel foam) without any conductive agents and binders [15,16,17,18,19]. However, other problems appeared, since the rigid structure can easily pierce the microporous membrane causing some safety hazards to the battery, although the metal substrate can provide good conductivity. 

Recently, high-flexibility self-supported electrodes have been obtained by replacing the metal conductive substrates with carbon fiber cloth. Mo et al. [20] reported on carbon-fiber cloth-supported 1D NiCo_2_O_4_ nanowire arrays (NCO@CFC) which exhibited superior Li^+^ storage capacity and rate capability than pure NiCo_2_O_4_. Zhao et al. [21] synthesized porous NiCo_2_O_4_ nanosheets directly on a carbon cloth substrate with surfactant polyvinylpyrrolidone (PVP), which presented good performance and superior rate capacity. However, the technical requirements and cost of the carbon cloth are higher than those for other materials; moreover, the nanostructured NiCo_2_O_4_ tended to agglomerate without surfactant (NH_4_F, C_6_H_12_N_4_, PVP) during the synthesis process and had poor electrochemical performance [22,23,24,25,26,27]. Therefore, it is necessary to find a low-cost material with good flexibility and electrical conductivity. Biomass is widely distributed almost everywhere in nature and its derived carbon materials have excellent electrical conductivity and structural stability. However, currently they cannot replace commercial graphite anodes because of the large differences between graphite and biomass-derived carbon. The plant fiber mask has excellent elasticity and is an artificial product of various plant fibers characterized by good electrical conductivity and flexibility after carbonization. It is an ideal conductive substrate material for energy storage electrodes. To the best of our knowledge, there are rare reports on this topic.

In this work, firstly, we developed a facile strategy to fabricate a film substrate from biomass-derived carbon fibers (BCFs) using a plant fiber mask as a carbon source and a two-step heat treatment. The obtained products possessed excellent electrical conductivity and flexibility. Further, two-dimensional ultrathin NiCo_2_O_4_ nanosheets were grown directly on the BCF substrate (denoted as NCO NSs/BCFs) without any surfactant. The prepared NCO NSs/BCFs composites exhibited significantly enhanced electrochemical behaviors as anode materials for LIBs in terms of specific capacity, rate performance, and cycling stability. The remarkable performances are mainly attributed to the synergistic properties of ultrathin 2D NiCo_2_O_4_ nanosheets and highly conductive and flexible BCF substrate, providing ready access pathways for rapid Li^+^ diffusion, excellent mechanical strength, and good stability. In addition, the constructed 2D NiCo_2_O_4_ nanosheet architecture not only reduces the electrical resistance effectively and increases the contact area between the electrode and the electrolyte, but also efficiently restricts the strain and volume change during the lithiation and delithiation reaction processes. Overall, this work highlights an interesting strategy for the rational preparation of low-cost, highly flexible, biomass-derived carbon fiber substrates and 2D ultrathin nanosheets for self-supported, flexible electrode materials (NCO NSs/BCFs) for high-performance LIBs.

## 2. Materials and Methods 

### 2.1. Synthesis of a Biomass-Derived Carbon-Fiber Film

All chemical reagents were purchased from Sinopharm Chemical Reagent Co., Ltd. (Shanghai, China), including Ni(NO_3_)_2_·6H_2_O, Co(NO_3_)_2_·6H_2_O, and urea. All the reagents were analytical-grade and were used without further purification. In addition, plant fiber masks (PFMs) were purchased from the market (Zhenjiang, Jiangsu, China). Firstly, the PFMs were cleaned by ultrasounds in deionized water and ethanol for 30 min and then dried in an oven. Secondly, in order to increase the stability of the structure, the dried PFMs were put into a tube furnace and heated at 240 °C for 3 h in air at a heating rate of 5 °C·min^−1^, after which they changed from white to light yellow. Then, the pre-oxidation PFMs were further heat-treated at different temperatures (600, 800, and 1000 °C) for 1 h in argon at a heating rate of 5 °C·min^−1^. Finally, the resultant black BCF film was cleaned ultrasonically in 2 M HCl to remove impurities, washed with deionized water, and dried at 80 °C overnight. According to the temperature of the heat treatment, the samples were simply indicated as BCFs-T (T = 600, 800, and 1000 °C).

### 2.2. Synthesis of NiCo_2_O_4_ NSs/BCF Composites 

In a typical process, 1 mmol Ni(NO_3_)_2_∙6H_2_O, 2 mmol Co(NO_3_)_2_∙6H_2_O, and 5 mmol urea were dissolved into 35 mL methanol to form a transparent pink solution. A piece of BCFs-800 was added and sonicated for 5 min, before transferring into a Teflon-lined stainless-steel autoclave (50 mL) at 120 °C for 6, 8, and 10 h. After hydrothermal growth, the BCFs encapsulated in the NiCo precursor were carefully washed several times with deionized water and ethanol. Finally, the samples were annealed at 300 °C for 2 h in air at a heating rate of 2 °C·min^−1^, resulting in the final NCO NSs/BCFs-t (t = 6, 8, and 10 h).

### 2.3. Characterization Methods 

The surface morphology of the composites was characterized by field-emission scanning electron microscopy (FE-SEM, JSM-6480, Japan), and their element distribution was measured by energy-dispersive spectrometry (EDS). X-ray photoelectron spectroscopy (XPS) using an ESCALAB 250Xi X-ray photoelectron spectrometer (MA, USA) with Al–Kα X-rays as the excitation source was applied to analyze the electronic structure. The crystal structure was analyzed by X-ray diffraction (XRD, XRD-6000, Japan) with Cu K*a* radiation (*β* = 0.15406 nm) operated at 40 KV and 30 mA at the 2θ range of 10–80°. Raman spectra were recorded by using a micro-Raman spectrometer (Invia Renishaw, UK) in the back-scattering geometry, with a 532 nm laser as the excitation source at room temperature.

### 2.4. Electrochemical Measurement

The electrochemical measurements were conducted in CR2025 coin-type half cells, which were assembled in an argon-filled glove box (H_2_O, O_2_ < 0.1 ppm). The biomass-derived carbon fiber (BCFs-T) film and NiCo_2_O_4_ NSs/BCFs composite (the mass loading of NiCo_2_O_4_ is over 1.3 mg·cm^−2^) directly acted as the working electrode without any conductive agent and binder. A metal lithium foil was employed as a reference and counter electrode, a microporous membrane (Gelgard 2500) was used as the separator, and the electrolyte was composed of 1 M LiPF_6_ in ethylene carbonate (EC)/dimethyl carbonate (DMC)/diethyl carbonate (DEC) (1:1:1, vol%). Cyclic voltammetry (CV) was performed using an electrochemical workstation (VSP-300, Bio-logic, France) with a potential window of 0.01–3.0 V, and electrochemical impedance spectroscopy (EIS) was performed on the electrochemical workstation over a frequency range from 10 mHz to 1 MHz at a potential amplitude of 5 mV. The charge–discharge, cycle, and rate performances were determined using a multichannel battery testing system (CT2001A, Land, Wuhan, China) in a voltage window of 0.01–3.0 V. 

## 3. Results and Discussion

As illustrated in Figure 1, 2D ultrathin NiCo_2_O_4_ nanosheets were grown vertically on a biomass-derived carbon fiber film (NCO NSs/BCFs). Firstly, the BCF substrate was obtained by two-step heat treatment, including pre-oxidation in air to improve the stability and high-temperature carbonization in argon to enhance the conductivity of the carbon material. Secondly, the NiCo_2_O_4_ precursor was synthesized by a hydrothermal method, described by the following Equations (1)–(3) [28,29]:6*Co*(*NH*_2_)_2_ → *C*_3_*H*_6_*N*_6_ + 6*NH*_3_ + 3*CO*_2_(1)
(2)$${\mathrm{NH}}_{3}+H_{2}O_{\;}\to \;NH_{4}^{+}+\;OH^{-}$$
(3)$$Ni^{2+}\;\;+\;2Co^{2+}\;+\;6OH^{-}\;\to \;{\mathrm{NiCo}}_{2}(\mathrm{OH}{)}_{6}$$

Finally, the NCO NSs/BCFs composite was obtained by further annealing in air at 300 °C for 2 h, which can be described by a simple oxidation process with Equation (4):2*NiCo*_2_(*OH*)_6_ + *O*_2_ → 2*NiCo*_2_O_4_ + 6*H*_2_*O*(4)

The morphology of the obtained materials was examined by SEM. Figure 2a shows a low-magnification SEM image of the BCF substrate, which displays a 3D network structure composed of intertwined carbon fibers with good flexibility. Further observation shows that the carbon fiber had a smooth surface and a diameter of about 5–6 μm (Figure 2b). Figure 3 are the SEM images of the composites NCO NSs/BCFs-6h (a1-a3), NCO NSs/BCFs-8h (b1-b3), and NCO NSs/BCFs-10h (c1-c3), respectively. Figure 3(a1–3a3) shows the surface morphology of NCO NSs/BCFs-6h, in which the biomass-derived carbon fibers were completely encapsulated in ultrathin nickel cobaltate nanosheets. Some microspheres between the carbon fibers (Figure 3(a1)) can be seen, which derived from the agglomeration of NiCo_2_O_4_ nanosheets. The higher-magnification SEM images in Figure 3(a2,a3) clearly display the NiCo_2_O_4_ nanosheets vertically grown on the carbon fiber to form the NCO NSs/BCFs composite. The 2D NiCo_2_O_4_ nanosheets were thin and uniform, and there was a large space in the 3D network composed of crumpled NiCo_2_O_4_ nanosheets. The NiCo_2_O_4_ microspheres disappeared as the hydrothermal time was prolonged to 8 h (NCO NSs/BCFs-8h). Most likely, the NiCo_2_O_4_ microspheres fell off from the carbon fibers when the NiCo_2_O_4_ nanosheets grew sufficiently and were removed by the subsequent cleaning process (Figure 3(b1)). Enlarged images of NCO NSs/BCFs-8h are shown in Figure 3(b2,b3), which show that NiCo_2_O_4_ nanosheets became thicker and larger compared to the previous sample. At the same time, the vacant volume between the 2D NiCo_2_O_4_ nanosheets further enlarged. This allowed not only to accommodate more electrolyte and improve Li^+^ diffusion efficiency, but also to efficiently limit the volume change and enhance the stability of the whole electrode material.

Figure 3(c1–c3) shows the morphology of NCO NSs/BCFs-10h at different resolutions. Many disorganized NiCo_2_O_4_ clusters in the NCO NSs/BCFs composite can be seen, which could increase the resistance and affect the charge transfer efficiency. Figure 3d shows the corresponding EDS element mapping of Ni, Co, and O for NCO NSs/BCFs-10h. The results demonstrated that NiCo_2_O_4_ nanosheets uniformly grew on the carbon fibers to form the NCO NSs/BCFs composite.

Figure 4a shows the Raman spectra of the BCF substrate. The two peaks located at 1350 and 1590 cm^−1^ correspond to the D and G band, respectively. The D band was caused by the sp^3^ disorders and defects in the graphitic structure, and the G band was attributed to the sp^2^ graphitic carbon atoms E_2g_ in-plane vibration. The ratio between the intensities of the D and G bands (I_D_/I_G_) reflected the defects of the carbon materials. The I_D_/I_G_ of BCFs-600, BCFs-800, and BCFs-1000 were 0.67, 0.86, and 0.94, respectively, which indicated that more defects were introduced by the high-heat treatment. It can be attributed to the pre-oxidation and carbonization processes occurring at high temperatures. These defects can provide a large number of active sites, which would be beneficial to the vertically growth of NiCo_2_O_4_ in the subsequent hydrothermal process [30,31]. The XRD of BCFs in Figure 4b shows a broadened peak at around 24°, which indicates the amorphous structure of BCFs [32,33]. As the hydrothermal time increased, the nickel cobaltate nanosheets grew and became thicker, therefore, the weaker diffraction peak of the amorphous carbon substrates for NCO NSs/BCFs-8h and NCO NSs/BCFs-10h almost disappeared. All the other diffraction peaks of the NCO NSs/BCFs composites were indexed as cubic spinel NiCo_2_O_4_ phase (JCPDS no. 20-0781), and the theoretical oxidation states of the transition metal (TM) ions were Ni^2+^ and Co^3+^ [34,35]. No residues and impurities were detected, indicating the high purity of the samples.

The elemental composition and oxidation state of the vertically grown NiCo_2_O_4_ nanosheets were further characterized by X-ray photoelectron (XPS). The survey spectra (Figure 4c) indicated the presence of C, O, Ni, and Co; the C element was derived from the BCFs substrate. Figure 4d shows the high-resolution Ni 2p spectrum of NCO NSs/BCFs. The Ni 2p was fitted with two spin-orbit doublets (Ni^2+^ and Ni^3+^) and two shakeup satellite (indicated as “Sat.”) by using a Gaussian fitting method. The binding energy of 873.8 and 855.8 eV are characteristic of 2P_1/2_ and 2P_3/2_ of Ni^2+^, respectively, and the binding energy of 872.3 and 854.1 eV are characteristic of 2P_1/2_ and 2P_3/2_ of Ni^3+^, respectively [36,37]. Similarly, the high-resolution Co 2p spectra of NCO NSs/BCFs are shown in Figure 4e. The binding energy of 780.1 and 795.2 eV are characteristic of 2P_1/2_ and 2P_3/2_ of Co^2+^, respectively, and the binding energy of 780.9 and 796.9 eV are characteristic of 2P_1/2_ and 2P_3/2_ of Co^3+^, respectively. Besides, the shakeup peaks at the binding energy of 804 eV can be regarded as “Sat.” [37,38,39]. The high-resolution O 1s spectra (Figure 4f) could be divided into three peaks at binding energies of 529.6 (O1), 531.4 (O2), and 532.3 (O3). Generally, O1 is typical of metal–oxygen bonds (Ni–O and Co–O), O2 is commonly associated with defects caused by low-oxygen coordination, and O3 is ascribed to physically/chemically adsorbed H_2_O on the surface [40,41]. The results showed that the chemical composition of vertically grown NiCo_2_O_4_ nanosheets included Co^2+^, Co^3+^, Ni^2+^, and Ni^3+^, in good agreement with other reports on NiCo_2_O_4_.

Figure 5a shows the initial CV curves corresponding to three cycles of the BCF substrate. In the discharge process, the shape peak at 0.01 V was due to the complex phase transition caused by Li^+^ embedded in the carbon materials. A small peak near 0.5 V appeared in the second cycle, possibly due to the reversible redox reaction between the oxygen-containing functional group on the surface of the carbon material and Li^+^ in the electrolyte, as well as to other reversible side reactions. During the charge process, wide peaks from 0.2 to 1.0 V were recorded and were attributed to Li^+^ extraction from the carbon layers or to defects. Figure 5b shows the CV curves of the NCO NSs/BCFs electrode in three cycles at a scan rate of 0.1 mV·s^−1^. The strong cathodic peak located around 0.7 V in the initial cycle corresponds to the reduction of NiCo_2_O_4_ to metallic Ni, Co, and Li_2_O (Equation (5)), which is an irreversible process [28]. The broad peak around 0.3 V which disappeared in the next cycles can be attributed to the formation of a solid electrolyte interface (SEI) film, which is the main reason for the irreversible capacity. During the charge process, two broad oxidation peaks were found at approximately 1.4 and 2.25 V, which were due to the lithium extraction process (Equations (6)–(8), from right to left) [28,42,43]. Two reduction peaks were observed at 0.95 and 1.2 V during the subsequent discharge process, which were related to the formation of NiO, CoO, and Co_3_O_4_, respectively (Equations (6)–(8), from left to right). More importantly, the subsequent cycle curves almost overlapped, which demonstrated good electrochemical reversibility and stability of the Li^+^ insertion and extraction processes.
(5)$${\mathrm{NiCo}}_{2}O_{4}\;+\;8Li^{+}\;+\;8e^{-}\;\to \;Ni\;+\;2Co\;+\;4{\mathrm{Li}}_{2}O$$
(6)$${\mathrm{Li}}_{2}O+\mathrm{Ni}\;⇌\;NiO+\;2Li^{+\;}\;+\;\;2e^{-}$$
(7)$$2{\mathrm{Li}}_{2}O+2\mathrm{Co}⇌\;2CoO+\;4Li^{+}\;\;+\;\;4e^{-}$$
(8)$${\mathrm{Li}}_{2}O+\;3CoO⇌\;{\mathrm{Co}}_{3}O_{4}\;+\;2Li^{+\;}\;+\;\;2e^{-}$$

Figure 5c shows the galvanostatic charge–discharge (GCD) profiles of the BCF electrode at 0.1 A·g^−1^. The specific capacities of the first discharge and charge were 895.8 and 473.4 mA·h·g^−1^, respectively, and the corresponding initial coulombic efficiency (ICE) was 53%. The profiles almost overlapped with subsequent cycles, which indicates the good stability of the BCF electrode. Similarly, Figure 5d shows the GCD profiles of the NCO NSs/BCFs electrode for the 1st, 10th, 20th, and 50th cycles at 0.1 A·g^−1^. The high charge and discharge capacities of the NCO NSs/BCFs electrode during the first cycle were 1733.8 and 2987.7 mA·h·g^−1^, respectively, and the ICE was 58%. The large irreversible capacity was due to the formation of a SEI film and to sde reactions in the first discharge process. Besides, the discharge capacities were 1349.2, 1280.3, and 1197.3 mA·h·g^−1^ for the 10th, 20th, and 50th cycle, respectively, demonstrating good cycle stability of the NCO NSs/BCFs electrode. At the current density of 0.1 A·g^−1^, BCFs-T were cycled up to 80 cycles, as shown in Figure 5e. The BCFs-800 and BCFs-1000 electrodes showed a better cycle performance than the BCFs-600 electrode with almost no capacity loss, and their capacities remained 276.6 and 268.1 mA·h·g^−1^ after 80 cycles, respectively. Cycle and rate performance are important parameters for the practical application of LIBs. The contribution of the carbon-fiber substrate capacity to NCO NSs/BCFs-T was about 280 mA·h·g^−1^. As shown in Figure 5f, the NCO NSs/BCFs-8h electrode displayed an outstanding cycle stability, with a specific capacity as high as 1128 mA·h·g^−1^ after 80 cycles, which was better than those of NCO NSs/BCFs-6h and NCO NSs/BCFs-10h. Also, the NCO NSs/BCFs-8h electrode exhibited an outstanding rate capability, delivering large reversible specific capacities of 1312.1, 1035.9, and 818.5 mA·h·g^−1^ at the current densities of 0.1, 0.5, and 1 A·g^−1^, respectively (Figure 5g). When the current density returned to 0.1 A·g^−1^ after the rate capacity test, the specific capacity still released 1247 mA·h·g^−1^. Figure 5h shows the long cycle tests and corresponding coulombic efficiency of the flexible NCO NSs/BCFs-8h electrode at a current density of 0.5 A·g^−1^. The specific capacity dropped slightly during the first few cycles, which can be attributed to the insufficient lithiation reaction at high current density. The capacity gradually rose with the increase of the cycle number and finally stabilized at about 1100 mA·h·g^−1^. The performances of various NiCo_2_O_4_-based materials are summarized and compared in the Appendix A, which shows that our obtained NCO NS/BCF composite has a great competitive advantage due to the electrode activation process. The excellent electrochemical performance of the NCO NSs/BCFs-8h electrode can be attributed to the unique 2D NiCo_2_O_4_ nanosheet structure and to the highly conductive BCF substrate.

EIS was further employed in the electrode reaction in the frequency range from 10 mHz to 1 MHz [44]. Nyquist plots of BCFs are shown in Figure 6a. In the high-frequency region, R_S_ is the internal resistance of the tested battery, which can be described by the intercept of the *X*-axis. The calculated values were 2.45, 2.1, and 1.71 Ω for BCFs-600, BCFs-800, and BCFs-1000, respectively, proving a small internal resistance. Similarly, Figure 6b shows the Nyquist plots (inset, equivalent circuit model) of the NCO NSs/BCFs electrode. Rct (semicircular diameter) represents the charge transfer resistance through the electrode–electrolyte interface. By contrast, the NCO NSs/BCFs-6h and NCO NSs/BCFs-8h electrodes with relatively low Rct exhibited better electrolyte wettability, which is related to the presence of the ultrathin NiCo_2_O_4_ nanosheets and to their ordered arrangement. CPE in Figure 6b is the constant phase element, and the inclined line in the low-frequency region corresponds to the Warburg impedance (Z_W_) during the diffusion of lithium inside the electrode material.

The flexible self-supported NCO NSs/BCFs electrode as an anode for LIBs is shown in Figure 7; its excellent electrochemical performance can be attributed to the following aspects:

(1) the BCF substrate has superior flexibility and mechanical strength; it provides not only good support for NCO NSs/BCFs composites, but also high-speed paths for electron transport thanks to the high conductivity of the carbon fibers.

(2) Vertically grown 2D NiCo_2_O_4_ nanosheets provide a large contact area between the electrode and the electrolyte, which shortens the ions/electrons transport distance. More importantly, the nanosheets structure can effectively limit the volume change during Li^+^ insertion and extraction and improve the stability of the electrode material.

(3) Compared with the conventional coated electrodes, a self-supported electrode without conductive agents and binders can effectively avoid electrode polarization problems.

## 4. Conclusions

In this work, a flexible BCF film was successfully prepared by a simple two-step thermal treatment including pre-oxidation and high-temperature carbonization. The BCF film had superior flexibility, good mechanical strength, and outstanding conductivity and provided more high-speed paths for electron transport. Subsequently, a self-supported NCO NSs/BCFs hybrid film was obtained by the vertically growth of NiCo_2_O_4_ nanosheets on the BCF film substrate via a hydrothermal process and annealing, in which ultrathin NiCo_2_O_4_ nanosheets were uniformly encapsulated on highly conductive carbon fibers to form a cross-linked network. In addition, the 2D NiCo_2_O_4_ nanosheets provided a large contact area between the electrode and the electrolyte, shortened the ions/electrons transport distance, and effectively limited the volume change during Li^+^ insertion and extraction. Benefiting from these interesting configurations, the self-supported NCO NSs/BCFs film was directly employed as an additive-free electrode for LIBs, which exhibited high specific capacity (1128 mA·h·g^−1^ after 80 cycles at 100 mA·g^−1^), favorable rate capability (818.5 mA·h·g^−1^ at 1000 mA·g^−1^), and cycling stability. This work provides new insights for the fabrication of low-cost self-supported electrode materials for a new generation of energy-storage devices. It is worth noting that this synthesis strategy for NCO NSs/BCFs composites can be extended to the preparation of other TMO-based self-supported electrode materials for use as high-performance energy-storage and conversion devices.

## Figures and Tables

**Figure 1 nanomaterials-09-01336-f001:**
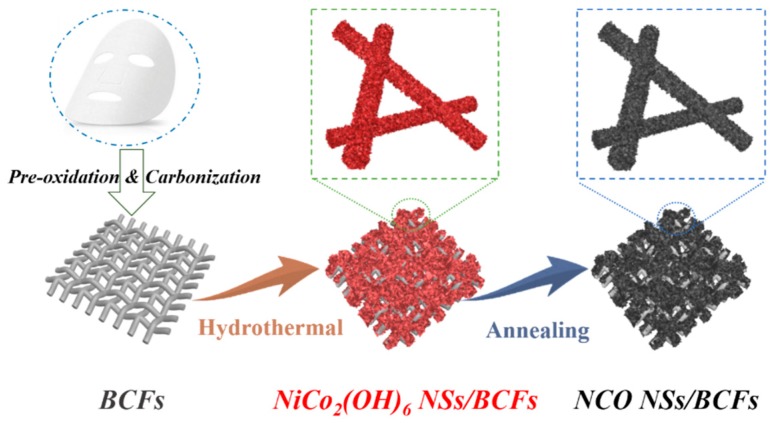
Schematic illustration of the synthesis of the composite consisting of NiCo_2_O_4_ nanosheets grown vertically on a biomass-derived carbon fiber film (NCO NSs/BCFs).

**Figure 2 nanomaterials-09-01336-f002:**
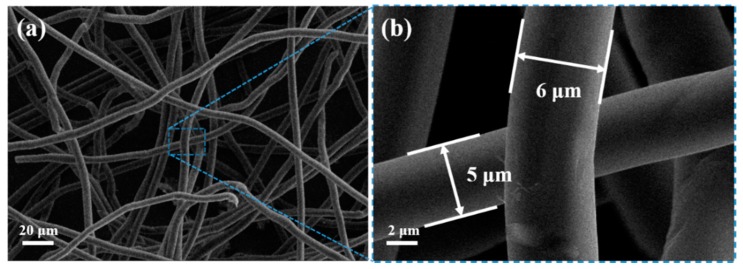
(**a**) and (**b**) SEM images of the BCF substrate.

**Figure 3 nanomaterials-09-01336-f003:**
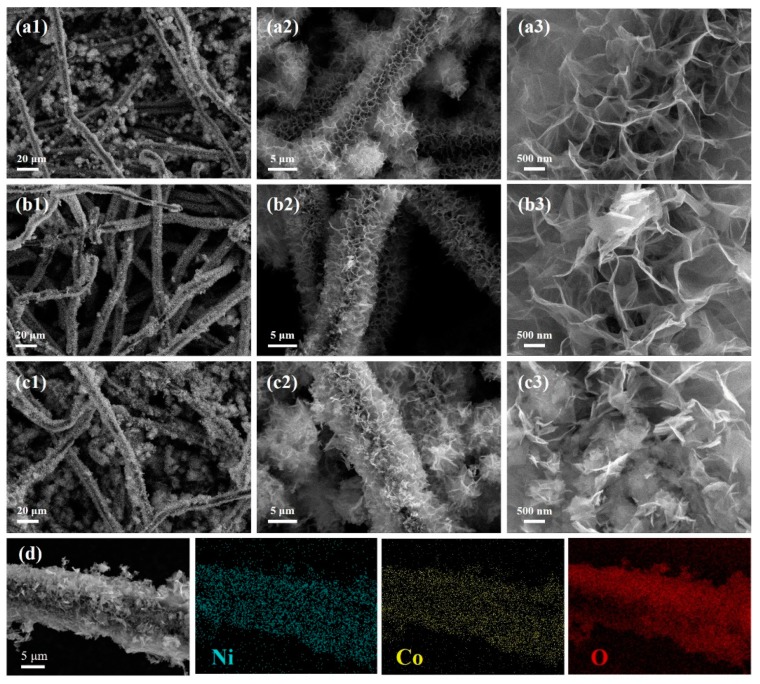
SEM images of the (**a1**–**a3**) NCO NSs/BCFs-6h; (**b1**–**b3**) NCO NSs/BCFs-8h; (**c1**–**c3**) NCO NSs/BCFs-10h; (**d**) EDS elemental mapping of Ni, Co, and O for NCO NSs/BCFs-10h.

**Figure 4 nanomaterials-09-01336-f004:**
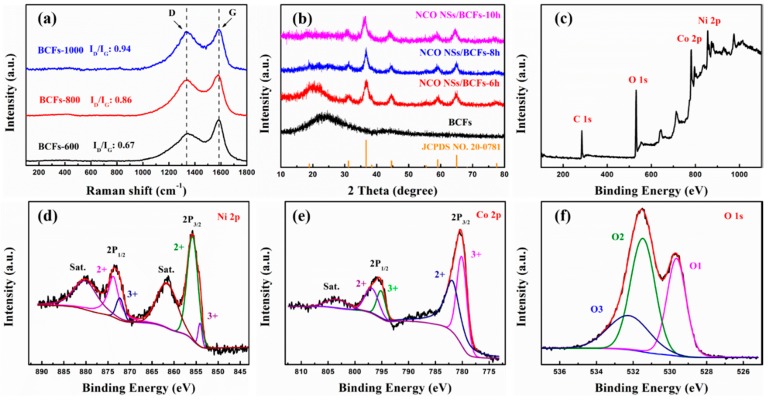
(**a**) Raman spectra of BCFs; (**b**) XRD pattern of NCO NSs/BCFs; (**c**) XPS survey spectra of NCO NSs/BCFs; High-resolution (**d**) Ni 2p; (**e**) Co 2p; (**f**) O 1s XPS spectrum of NCO NSs/BCFs.

**Figure 5 nanomaterials-09-01336-f005:**
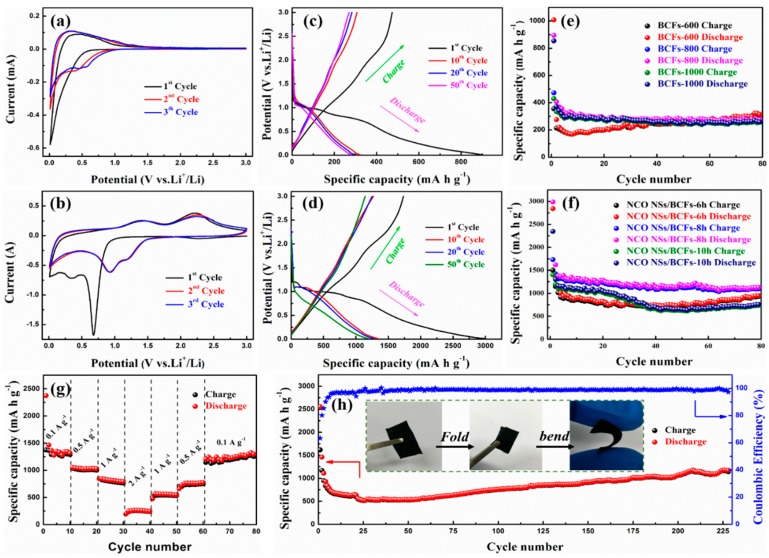
CV curves of (**a**) BCFs and (**b**) NCO NSs/BCFs; galvanostatic charge–discharge (GCD) profiles of (**c**) BCFs and (**d**) NCO NSs/BCFs; cycle performance of (**e**) BCFs and (**f**) NCO NSs/BCFs composite; (**g**) rate performance; (**h**) long cycle performance and corresponding coulombic efficiency of NCO NSs/BCFs-8h.

**Figure 6 nanomaterials-09-01336-f006:**
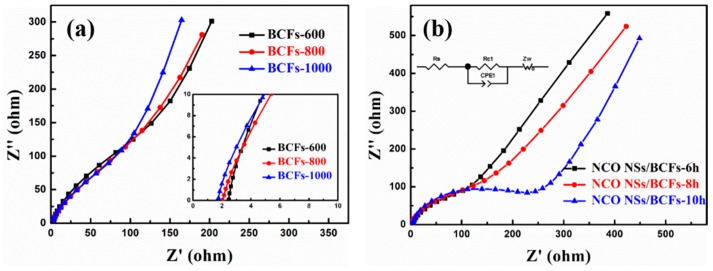
Nyquist plots of (**a**) BCFs; (**b**) NCO NSs/BCFs composite.

**Figure 7 nanomaterials-09-01336-f007:**
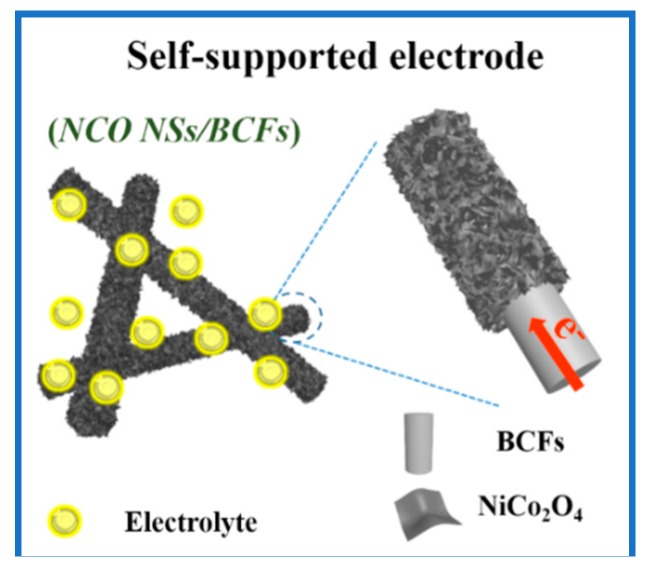
Schematic diagram of the self-supported NCO NSs/BCFs electrode.

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
