# Peer review of "Vertically Aligned NiCo2O4 Nanosheet-Encapsulated Carbon Fibers as a Self-Supported Electrode for Superior Li+ Storage Performance"

_nanomaterials, 2019, doi:10.3390/nano9091336_

Round 1

Reviewer 1 Report

The authors have synthesised a metal oxide composite and have used it as an anode material in LIB. They have characterised the material thoroughly and investigated its battery performance. The paper can be published with some very minor corrections.

Few additional references need to be added for transition metal oxides as anode material in the introduction (Mat. Chem. Front., 2017, 1, 2213; J. Electrochem. Soc., 2017, 164, D597; Nanoscale, 2012, 4, 2526)

Author Response

Reply to Reviewer #1

Overall assessment - The authors have synthesised a metal oxide composite and have used it as an anode material in LIB. They have characterised the material thoroughly and investigated its battery performance. The paper can be published with some very minor corrections.

Comment #1: Few additional references need to be added for transition metal oxides as anode material in the introduction (Mat. Chem. Front., 2017, 1, 2213; J. Electrochem. Soc., 2017, 164, D597; Nanoscale, 2012, 4, 2526).

Our response: We thank the reviewer for this appropriate comment, and we agree that some related citations are required. Following reviewer’s suggestion, we carefully read the literatures and cited some of suggested literatures to support the application of transition metal oxides in electrochemical energy derives.

Mat. Chem. Front., 2017, 1, 2213; (cited in Ref 8.)

J. Electrochem. Soc., 2017, 164, D597; (cited in Ref 3.)

Nanoscale, 2012, 4, 2526; (cited in Ref 14.)

Reviewer 2 Report

Authors fabricated biomass-derived carbon fibers film as the substrate to synthesize 2D NiCo2O4 nanosheets as the flexible electrode for LIBs.

I have some questions as follow:

The text of “PEMs” in Line 91 on page 2 should be a wrong typing. The author should provide more information about the PFMs, Ni(NO3)26H2O, Co(NO3)2·6H2O, and urea which purchased from the market. For example, which companies fabricate those materials? Why oxidation needs for PFMs? Why 240 degree C used for oxidation? How to confirm the oxidation level? In Fig. 4b, why the peak of BCFs at 24 degree shift to around 20 degree of sample NCO NSs/BCFs-6h? Why the peaks of BCFs at 24 degree almost disappeared for samples NCO NSs/BCFs-8h and NCO NSs/BCFs-10h? In Fig. 4c, authors marked the peaks of C1s, O1s, Co 2p, and Ni 2p. what are the other peaks in Fig. 4c ? In Fig. 5h, the authors showed the image of sample bending. How about the stretchability?

Author Response

Reply to Reviewer #2

Overall assessment - Authors fabricated biomass-derived carbon fibers film as the substrate to synthesize 2D NiCo2O4 nanosheets as the flexible electrode for LIBs. I have some questions as follow:

Comment #1: The text of “PEMs” in Line 91 on page 2 should be a wrong typing. The author should provide more information about the PFMs, Ni(NO3)2·6H2O, Co(NO3)2·6H2O, and urea which purchased from the market. For example, which companies fabricate those materials?

Our response: We thank the reviewer’s valuable comment and added some details in the manuscript. Such as “All chemical reagents were purchased from Sinopharm Chemical Reagent Co., Ltd., including Ni(NO3)2·6H2O, Co(NO3)2·6H2O, and urea. All the reagents were analytical grade and were used without further purification. In addition, Plant fiber marks (PFMs) were purchased from the market.”

Comment #2: Why oxidation needs for PFMs? Why 240 degree C used for oxidation? How to confirm the oxidation level?

Our response: We thank the reviewer’s professional comments. Low temperature oxidation treatment is a common method for pretreating carbon fibers, and the etching effect of air can increase the defects on the surface of carbon fiber, which is beneficial to the hydrothermal growth of nickel cobaltate precursor (NiCo2(OH)6). The choice of oxidation temperature is based on the experimental process of the relevant literature, and the color will be light yellow after oxidation treatment. We addressed reviewer’s concerns in manuscript as: “Secondly, in order to increase the stability of the structure, the dried PFMs were put into a tube furnace and heated at 240 oC for 3 h in air under a heating rate of 5 oC min-1, and the PFMs changed from white to light yellow.”

Comment #3: In Fig. 4b, why the peak of BCFs at 24 degree shift to around 20 degree of sample NCO NSs/BCFs-6h? Why the peaks of BCFs at 24 degree almost disappeared for samples NCO NSs/BCFs-8h and NCO NSs/BCFs-10h?

Our response: We thank the reviewer’s comment. The peak of NCO NSs/BCFs-6h at about 20 degree may be due to the side reactions on the surface of carbon fiber during hydrothermal process. As the hydrothermal time increases, the nickel cobaltate nanosheets increase and thicken, and the diffraction peaks of the amorphous carbon substrates is weak, so that the peak at 24 degree almost disappeared of NCO NSs/BCFs-8h and NCO NSs/BCFs-10h. We addressed reviewer’s concerns in manuscript as: As the hydrothermal time increases, the nickel cobaltate nanosheets increase and become thicker, therefore, the weaker diffraction peak of the amorphous carbon substrates for NCO NSs/BCFs-8h and NCO NSs/BCFs-10h almost disappeared.

Comment #4: In Fig. 4c, authors marked the peaks of C1s, O1s, Co 2p, and Ni 2p. what are the other peaks in Fig. 4c?

Our response: We thank the reviewer for this kind comment. We have consulted a large number of related literatures and found that similar peaks appear in the XPS of nickel cobaltate, which may be related to the formation of nickel cobaltate or testing instruments. As this research carry on, we hope to figure out in the next stage for this project.

Comment #5: In Fig. 5h, the authors showed the image of sample bending. How about the stretchability?

Our response: We thank the reviewer’s valuable comment. The plant fiber masks (PFMs) is similar to cotton fabric. After high-temperature carbonization, the adhesion between the fibers is poor, so the samples strechability are not good.

Reviewer 3 Report

In this work, the authors investigated electrochemical properties of the vertical aligned NiCo2 nanosheets encapsulated carbon fibers as a self-supported electrode for Li+ storage. The manuscript is written well, and the concept is very interesting. However, I believe some data are needed or updated to make the proposed concept more convincingly.

The author claimed that the biomass originated C is very cost-saving and practical. However, the information of the practical application of the biomass C compared to that of graphite is not sufficient. The authors should provide the practical merits of the C more specifically in the introduction. The mass loading of active materials should be indicated in the experimental part, since it is an important value, which is closely related to the electrochemical performances. The author should calculate the capacity contribution of C matrix to the NiCo2/C The detail morphologies of the composite after cycling have to be provided to prove the structure stability of the composite. It is necessary to provide the rate property of the cell above 5 A g-1. The following relevant papers for the flexible freestanding anode materials might be referred in this manuscript: S. H. Oh, O. H. Kwon, Y. C. Kang, J. K. Kim*, J. S. Cho, Journal of Materials Chemistry A 7 (2019) 12480-12488.

Author Response

Reply to Reviewer #3

Overall assessment - In this work, the authors investigated electrochemical properties of the vertical aligned NiCo2O4 nanosheets encapsulated carbon fibers as a self-supported electrode for Li+ storage. The manuscript is written well, and the concept is very interesting. However, I believe some data are needed or updated to make the proposed concept more convincingly.

Comment #1: The author claimed that the biomass originated C is very cost-saving and practical. However, the information of the practical application of the biomass C compared to that of graphite is not sufficient. The authors should provide the practical merits of the C more specifically in the introduction.

Our response: We thank the reviewer’s valuable comment. The advantages of biomass carbon materials are low cost and wide distribution; however, they have not been applied to commercial battery instead of graphite anodes. We aim to study 2D nickel cobaltate materials with biomass carbon as the base material. We addressed reviewer’s concerns in manuscript as: Biomass is widely distributed almost everywhere in nature and its derived carbon materials have excellent electrical conductivity and structural stability. However, it is currently impossible to replace commercial graphite anode due to the large differences in biomass-derived carbon.

Comment #2: The mass loading of active materials should be indicated in the experimental part, since it is an important value, which is closely related to the electrochemical performances.

Our response: We thank the reviewer’s timely comment. We added the mass loading of active materials in the experimental part of the manuscript. We addressed reviewer’s concerns in manuscript as: The biomass-derived carbon fiber (BCFs-T) film and NiCo2O4 NSs/BCFs composite (the mass loading of NiCo2O4 is over 1 mg) was directly acted as the working electrode without any conductive agent and binder.

Comment #3: The author should calculate the capacity contribution of C matrix to the NiCo2O4/C. The detail morphologies of the composite after cycling have to be provided to prove the structure stability of the composite. It is necessary to provide the rate property of the cell above 5 A g-1.

Our response: We thank the reviewer’s valuable comment. The carbon of all NiCo2O4/C composites is BCF-800 obtained by carbonization at 800 oC, and its electrochemical properties have been studied before, so we can reasonably think that the capacity contribution of C in NiCo2O4/C composites is about 270 mAh g-1. Unfortunately, we are temporarily unable to provide an SEM image of NiCo2O4/C electrode after cycling. We tried to test the performance of the NiCo2O4/C electrode material at a current density of 5 A g-1, but the results were not satisfactory. We addressed reviewer’s concerns in manuscript as: The carbon fiber substrate capacity contribution in NCO NSs/BCFs-T is about 280 mAh g-1. As shown in Fig. 5f, the NCO NSs/BCFs-8h electrode delivers outstanding cycle stability with a specific capacity as high as 1128 mA h g-1 after 80 cycles, which is better than that of NCO NSs/BCFs-6h and NCO NSs/BCFs-10h.

Comment #4: The following relevant papers for the flexible freestanding anode materials might be referred in this manuscript: S. H. Oh, O. H. Kwon, Y. C. Kang, J. K. Kim*, J. S. Cho, Journal of Materials Chemistry A 7 (2019) 12480-12488.

Our response: We thank the reviewer’s comment. The paper has been cited as Ref 22.

Reviewer 4 Report

This paper reports the electrochemical properties of vertically aligned NiCo2O4 nanosheets encapsulated carbon fibers as electrode for lithium batteries. The NiCo2O4@C composite synthesized by hydrothermal method displays good rate capability of 818.5 mAh g−1 at 1000 mA g−1. I consider that this paper is worthy to be published after major revision addressing the following issues.

(title) it should be LI<superscript>+ Is this composite electrode considered as conversion electrode? If so, the Eqs. 5-8 should be discussed accordingly. The electrode loading is missing, please indicate. What was the potential applied in EIS experiments? Line 181: please rewrite the sentence 6: the theoretical oxidation states of TM ions in the spinel structure should be documented Line 257: the authors evoked the charge transfer resistance but did not quantify. A comparison between the two types of electrode could be welcome. References: it should more comfortable for the reader having the title of the articles. Overall, a general discussion is missing. The authors could compare their data with the results from the literature as numerous studies have been devoted to the synthesis and characterization of NiCo2O4 as an anode material for Li-ion batteries.

Author Response

Reply to Reviewer #4

Overall assessment - This paper reports the electrochemical properties of vertically aligned NiCo2O4 nanosheets encapsulated carbon fibers as electrode for lithium batteries. The NiCo2O4@C composite synthesized by hydrothermal method displays good rate capability of 818.5 mAh g−1 at 1000 mA g−1. I consider that this paper is worthy to be published after major revision addressing the following issues.

Comment #1: (title) it should be LI<superscript>+ Is this composite electrode considered as conversion electrode? If so, the Eqs. 5-8 should be discussed accordingly.

Our response: We thank the reviewer’s reasonable comment. Nickel cobaltate is a typical conversion electrode material, and the redox peaks can be found in the CV curves of Fig. 5b. Equations 5-8 have been discussed in the manuscript. We addressed reviewer’s concerns in manuscript as: The strong cathodic peak located around 0.7 V in the initial cycle is corresponding to the reduction of NiCo2O4 to metallic Ni, Co and Li2O (Equation 5), which is an irreversible process [28]. A broad peak around 0.3 V disappeared in the next cycles can be attributed to the formation of solid electrolyte interface (SEI) film, which are the main reason for the irreversible capacity. During the charge process, two broad oxidation peaks were found at approximately 1.4 and 2.25 V, which were due to the lithium extraction process (Equations 6-8, from right to left) [28, 42, 43]. There are two reduction peaks at 0.95 and 1.2 V during the subsequent discharge process, which are related to the formation of NiO, CoO and Co3O4, respectively (Equations 6-8, from left to right).

Comment #2: The electrode loading is missing, please indicate. What was the potential applied in EIS experiments?

Our response: We thank the reviewer’s comment. We added the mass loading of active materials in the experimental part of the manuscript and rewritten the EIS experimental parameters. We addressed reviewer’s concerns in manuscript as: The biomass-derived carbon fiber (BCFs-T) film and NiCo2O4 NSs/BCFs composite (the mass loading of NiCo2O4 is over 1 mg) was directly acted as the working electrode without any conductive agent and binder.

Comment #3: Line 181: please rewrite the sentence 6: the theoretical oxidation states of TM ions in the spinel structure should be documented.

Our response: We thank the reviewer for this kind comment. We addressed reviewer’s concerns in manuscript as: All of other diffraction peaks of NCO NSs/BCFs composite are indexed to the be the cubic spinel NiCo2O4 phase (JCPDS no. 20-0781) and the theoretical oxidation states of TM ions are Ni2+ and Co3+.

Comment #4: Line 257: the authors evoked the charge transfer resistance but did not quantify. A comparison between the two types of electrode could be welcome.

Our response: We thank the reviewer’s reasonable comment. The charge transfer resistance of NCO NS/BCF-T can be clearly found by the semicircular diameter in Fig. 6b, the Rct values of NCO NS/BCF-6h, NCO NS/BCF-8h and NCO NS/BCF-10h are 103.5, 110.3 and 142.8 Ω, respectively. The first two have smaller Rct, indicating faster charge transfer efficiency.

Comment #5: References: it should more comfortable for the reader having the title of the articles. Overall, a general discussion is missing. The authors could compare their data with the results from the literature as numerous studies have been devoted to the synthesis and characterization of NiCo2O4 as an anode material for Li-ion batteries.

Our response: We thank the reviewer’s objective comment. We summarize and compare the performances of various NiCo2O4-based electrodes, which are shown in supporting information Table S1.

Table S1 Lithium storage performance of various NiCo2O4-based materials.

Material

Current density

(mA g-1)

Specific capacity

(mAh g-1)

reference

NiCo2O4

100

1175.9

1

NiCo2O4/C

500

1100

2

NiCo2O4 microrods

100

857.6

3

NiCo2O4 nanowires

4000

507

4

NiCo2O4 microspheres

100

1167

5

NiCo2O4/graphene nanosheets

100

1216

6

NiCo2O4 nanowires

200

976

7

NiCo2O4/C Ni Foam

100

1298

8

NiCo2O4 nanosheets

500

1687.6

9

NiCo2O4/C nanoparticles

100

1092

10

NiCo2O4 /CNTs

500

840

11

NiCo2O4/C nanowires

500

1012

12

NiCo2O4

Carbon Fiber Cloth

100

799

13

NiCo2O4 nanosheets

Plant Carbon Fiber

100

1128

This work

Round 2

Reviewer 2 Report

The revision is much better, I recommend to accept.

Author Response

Overall assessment - The revision is much better, I recommend to accept.

Our response: We thank the reviewer for your recognition of this article.

Reviewer 4 Report

The authors have partially satisfied the queries of the reviewer. I consider that several issues must be addressed before final acceptance.

The mass loading of an electrode is a very important parameter. It should be expressed in g/cm2. Images of the EDS mapping of Ni and Co are very dark. Please modify the contrast. (XPS section) The authors have well documented the theoretical oxidation states of TM ions. But experiments show a rather complex situation. In line 207, it is written “NiCo2O4 nanosheets contain Co2+, Co3+, Ni2+ and Ni3+” without discussion. Please explain clearly the oxidation state in NiCo2O4 and quantify the amount of each species. The article titles of the cited references are missing. It should be more comfortable for the reader to have it. Please complete.

Author Response

We thank the reviewer's valuable comment and point to point responses have been listed in the attachment.
